# Comparison of the Efficiency of Hetero- and Homogeneous Catalysts in Cellulose Liquefaction

**DOI:** 10.3390/ma16186135

**Published:** 2023-09-09

**Authors:** Paulina Kosmela, Paweł Kazimierski

**Affiliations:** 1Department of Polymer Technology, Chemical Faculty, Gdansk University of Technology, G. Narutowicza Str. 11/12, 80-233 Gdansk, Poland; paulina.kosmela@pg.edu.pl; 2Institute of Fluid Flow Machinery, Polish Academy of Sciences, Fiszera Str. 14, 80-231 Gdansk, Poland

**Keywords:** biomass, solvothermal liquefaction, bio-polyols, heterogeneous catalyst

## Abstract

Biomass liquefaction is a well-known and extensively described process. Hydrothermal processes are well understood and can be used in the fuel industry. The use of organic solvents can result in full-fledged products for use in the synthesis of polyurethanes. The plastics industry, including polyurethanes, is targeting new, more environmentally friendly solutions. One of these is the replacement of petrochemical polyols with compounds obtained from renewable sources. It is common in biomass liquefaction to use sulfuric acid (VI) as a catalyst. The purpose of the present study was to test the effectiveness of a heterogeneous catalyst such as Nafion ion-exchange resin on the cellulose liquefaction process. The results obtained were compared with the bio-polyols obtained in a conventional way, using a homogeneous catalyst (sulfuric acid (VI)). Depending on the catalyst used and the temperature of the process, bio-polyols characterized, among other things, by a hydroxyl number in the range of 740–400 mgKOH/g were obtained. The research provides new information on the possibility of using heterogeneous catalysts in cellulose liquefaction.

## 1. Introduction

A lot of scientific research is currently focused on the development of innovative technologies that can replace the conventional use of fossil fuel resources. One of the problems widely described in the literature is the replacement of petrochemical polyols with compounds obtained using renewable resources. As indicated in other works, vegetable oils [1,2,3] and biomass liquefaction [4,5,6] are most often used to obtain bio-polyols. Liquefaction belongs to one of the methods of the thermochemical conversion of biomass, among which we can also distinguish pyrolysis, combustion, or gasification [7,8].

The liquefaction process uses a variety of biomasses including wood [9], bamboo [10], straw [11], or biomass from water environments [12]. Due to the various possibilities for processing biomass, the end products resulting from its processing are used in various industries. In the process of the liquefaction of biomass, its type and particle size are of great importance, but the most important is the choice of other substances, including solvents and catalysts. The properties of the final products of the liquefaction process are also influenced by the conditions of the process. Physical factors such as temperature, pressure, mass ratio of components, catalyst concentration, and heating rate play an important factor [13,14,15].

Another very important component during liquefaction is the type of catalyst. More popular are homogeneous catalysts. Organic and inorganic acids, bases, and salts are used. The literature is familiar with the use of acidic catalysts (sulfuric acid, orthophosphoric acid, hydrochloric acid, and p-toluenesulfonic acid), basic catalysts (sodium hydroxide), and metallic salts. The use of strong acids can cause corrosion of the apparatus and contribute to greater environmental pollution compared to basic catalysts [16,17,18]. The use of heterogeneous catalysts in the liquefaction process has not been described in the literature. The use of heterogeneous catalysts will contribute to the elimination of the neutralization process, which is necessary when using homogeneous catalysts. By eliminating this step, liquefaction can become a more environmentally friendly process. The search for more efficient catalysts that can be recycled is an important part of “green chemistry”. Heterogeneous catalysts include polymeric perfluorohexanesulfonic acids (Nafion^®^, Amberlyst, among others), zeolites (zeolite HZSM-5 and zeolite Y), heteropolyacids (phosphotungstic acid and silicotungstic acid), tungstated zirconium oxide, and sulfated zirconium oxide.

Heterogeneous catalysts are widely used in chemical conversion. Zeolites have a porous structure and are aluminosilicate minerals. They have found applications in hydrothermic biomass liquefaction (HTL), among others. Chemn et al. [19] demonstrated that zeolites can be successfully used as catalysts for the HTL of pine sawdust. Zeolite HZSM-5 exhibits a high content of Brønsted acid sites, which promote dehydrogenation or cyclization, which favorably affects the hydrocarbon content of the final product. Another example is heteropolyacids. They are formed by condensing molecules of various inorganic acids, such as silicic acid and phosphoric acid. They find use, for example, as catalytic centers. An example of a Keggin-type heteropolyacid is phosphotungstic acid, which has found use in liquefaction using supercritical ethanol, among other applications. The high acidity of heteropolyacids makes them good catalysts for liquefaction, the efficiency of which largely depends on the strength of the acid used. The use of heteropolyacids carries many advantages, including low toxicity, environmental friendliness, availability, and the ability to dissolve biomass in polar solvents [20].

An example of a polymeric perfluororesin sulfonic acid is Nafion, which chemically is a copolymer of tetrafluoroethene (a Teflon monomer) and a perfluorinated oligovinyl ether terminated with a strongly acidic sulfonic residue. It has found widespread use in fuel cells, electrochemical equipment, water electrolysis, and acts as a superacid catalyst for high-value chemicals, among other applications [21]. Nafion is a superacid that can potentially be used as a catalyst in organic synthesis, for example, in alkylation, isomerization, oligomerization, or esterification reactions.

The bio-polyols obtained via liquefaction are widely described in the literature. Due to the difficulties in determining the chemical structure of bio-polyols, most works focus on describing the basic properties, i.e., the hydroxyl number or water content, and on the use of liquefaction products in polyurethane materials. Due to their extended structure, bio-polyols have found applications in rigid polyurethane foams [22,23] and cross-linked polyurethane resins [24,25].

This paper presents a comparison of the properties of the obtained bio-polyols using two types of catalysts. The homogeneous catalyst was the most commonly used sulfuric acid during liquefaction, and the heterogeneous catalyst was a polymeric perfluorohexanesulfonic acid with the trade name Nafion NR40. Of the various heterogeneous catalysts, Nafion NR40 was chosen due to the lack of literature reports on its use in biomass liquefaction. Cellulose was liquefied using solvents in the form of a mixture of glycerol and PEG400 in a 50/50 mass ratio. The obtained bio-polyols were characterized by analyzing the amount of hydroxyl groups, biomass conversion, determining oxidative stability, thermal stability, analyzing gas products via thermogravimetric analysis, determining chemical structure using the NMR technique, and determining molecular weight using the GPC technique. In addition, a mass balance of the liquefaction was presented to determine the efficiency of the process.

## 2. Experimental

### 2.1. Materials

Bio-polyols were obtained by liquefying cellulose using a mixture of solvents. The cellulose was supplied from POCH S.A. (Gliwice, Poland) in the form of cotton linters. Before the liquefaction process, the cellulose was dried at 105 °C for 24 h. The liquefaction solvent was a mixture (50/50 parts by weight) of glycerol and poly(ethylene oxide) with a molecular weight of 400 g/mol (PEG400). Glycerol (LOH ≈ 1312 ± 42 mgKOH/g) was supplied by Prochemica (Katowice, Poland) and PEG400 (LOH ≈ 286 ± 8 mgKOH/g) was supplied by PCC Rokita S.A. (Brzeg Dolny, Poland). Two types of catalysts were used during liquefaction. The homogeneous catalyst was sulfuric acid (VI) (95% solution) supplied by Avantor Performance Materials Poland S.A (Gliwice, Poland), and the heterogeneous catalyst was Nafion NR40 from DuPont Fuel Cells (Wilmington, NC, USA). Granular potassium hydroxide, supplied by Avantor Performance Materials Poland S.A (Gliwice, Poland), was used to neutralize the obtained bio-polyols.

### 2.2. Preparation of Bio-Polyols

Cellulose liquefaction was carried out in accordance with previous works [25,26,27]. The liquefaction process with hetero- and homogeneous catalysts looked similar. In the case of using sulfuric acid (VI) as a catalyst, a second synthesis step involving the neutralization of the bio-polyol was necessary. The liquefaction was carried out under atmospheric pressure in a reactor equipped with a mechanical stirrer, thermometer, and distillation cooler. Cellulose was placed in the reactor and solvent (glycerol/PEG400 mixture) was added. The mass ratio of cellulose to solvents was 1/10. The amount of catalyst was 3 (homogeneous catalyst) or 6% *w*/*w*. (heterogeneous catalyst) in relation to the solvents. The quantity of each component is shown in Table 1. Liquefaction was carried out for 6 h at different temperatures (130, 150, and 170 °C). To determine the liquefaction efficiency, a sample was taken every hour for analysis. During synthesis with sulfuric acid (VI), it was necessary to neutralize the reaction mixture. A specified amount of potassium hydroxide (in granule form) was added to the reactor, while neutralizing and drying the bio-polyol under reduced pressure for 2 h. Possible reactions occurring during liquefaction are shown in Figure 1.

### 2.3. Characterization

The hydroxyl number was determined according to PN-93/C-89.052/03 [28]. The sample was placed in flasks and 5 mL of the acellular mixture was added. Then, the whole was placed in a water bath for 30 min. After 30 min, 1 mL of pyridine was added, and after another 10 min, 50 mL of distilled water was added via a reflux condenser. The mixture was titrated with 0.5 M of the potassium hydroxide solution in the presence of phenolphthalein. For each sample, the content of hydroxyl groups was calculated according to Equation (1):(1)$$HV=\frac{56.1\;*\;\left(V_{2}-V_{1}\right)\;*\;C_{KOH}}{m}$$
where *V*_2_ [cm^3^] is the volume of titrant required for the titration of the blank solution, *V*_1_ [cm^3^] is the volume of titrant required for the titration of a sample, *C_KOH_* [M] is the molarity of the titrant, and *m* [g] is the mass of the sample.

The biomass conversion wattage was determined via the solid residue after dissolving the bio-polyol in ethanol. The sample was placed in a flask and dissolved with excess ethanol. The whole mixture was stirred for 1 h, until the sample was completely dissolved. The mixture was then filtered under reduced pressure using filter paper. The blotting paper was weighed before and after the test, and the biomass conversion was determined according to Equation (2).
(2)$$BC\;\left[\%\right]=100\%-\frac{m}{m_{0}}\;*\;100\%$$
where *m* [g] is the weight of the residual biomass and *m*_0_ [g] is the initial weight of the biomass.

The viscosity values of the prepared polyols were determined using an R/S Portable rheometer. The analysis was conducted at 25 °C for the shear rates varying from 1 to 100 s^−1^. The obtained results were analyzed by employing Rheo 3000 version 1.2 computer software.

The chemical structure was determined via nuclear magnetic resonance (NMR). ^1^H and ^13^C NMR analyses were carried out to determine the structure of the bioplastics. NMR spectra were recorded on a Varian Gemini 500 MHz spectrometer (Palo Alto, CA, USA). Chemical shifts were given in ppm relative to the residual solvent peak of DMSO-d6 = 2.49 ppm for ^1^H and 39.5 ppm for ^13^C.

The weight average molecular weight (Mw) and the number average molecular weight (Mn) were determined using the technique of size exclusion chromatography (HPLC-SEC). A Shimadzu Prominence GPC System liquid chromatograph was used, equipped with an LC-20AD pump, a DGU-20A3 Degazer, a SIL-20A Autosampler, a CTO-20A thermostat, a RID-10A refractometer detector, a CBM-20A interface, and a computer with LabSolutions software (Ezchrom 3.2). Column: two SDVB-based packed columns connected in series, 300 × 8 mm; eluent: DMF, eluent flow rate: 0.75 mL/min; temperature: 35 °C; dispensed sample volume: 50 uL; sample concentration: 0.01 g/mL; sample solvent: eluent; and detection: RI. Mass calibration against PEG/PEO. Molecular masses above 900th Da are extrapolated values. Exclusion mass is approximately 4 million Da.

The thermogravimetric analysis (TGA) was performed on a NETZSCH TG 209 apparatus (Selb, Upper Franconia, Germany) using 10 mg samples within the temperature range of 40–600 °C under a nitrogen atmosphere and at a heating rate of 10 °C/min.

Bio-poly degradation products were analyzed using the FTIR spectroscope Nicolet iS10 from Thermo Scientific (Waltham, MA, USA). The test was carried out for 4 mg of bio-polyol samples at a rate of 20 °C/min, in a nitrogen gas atmosphere over a range of temperatures from 25 °C to 700 °C.

The temperature of the onset of oxidation was determined using a NETZSCH DSC 204 F1 differential scanning calorimeter. Bio-polyol samples were placed in an open aluminum crucible and heated at 10 °C/min in an oxygen atmosphere over a temperature range of 25 °C to 700 °C. Using the NETZSCH Proteus Thermal Analysis software v.8.0.2., the oxidation onset temperature (OOT) was determined.

## 3. Results and Discussion 

### 3.1. Physicochemical Properties

Biomass liquefaction is a very complex process. In addition to liquefaction, other reactions can occur, e.g., condensation of the solvent, as confirmed in another paper [29]. In order to study the course of the reaction, the hydroxyl number, the conversion of biomass was determined during the study by taking a sample every hour. The liquefaction process was carried out at three different temperatures, and the bio-polyols were named based on the temperature of the process catalyst. For example, the bio-polyol obtained at 130 °C with a homogeneous catalyst was named 130_H_2_SO_4_. The results of the hydroxyl number determinations and the biomass conversion over time are shown in Figure 2.

For the bio-polyols obtained with the homogeneous catalyst, the HV and BC values for 8 h are additionally presented, as these bio-polyols were additionally neutralized and dried. Analyzing the results, it can be seen that in all cases, the values of the hydroxyl number decrease and biomass conversion increase over time. Significantly lower hydroxyl number values and higher biomass conversion values were achieved by running the process with sulfuric acid (VI) as a catalyst. This demonstrates the higher reactivity of this catalyst compared to the heterogeneous catalyst. As the temperature of the liquefaction process increases, lower HV values and higher BC values are observed regardless of the type of catalyst. In the case of reactions using sulfuric acid (VI), the greatest reaction progress is observed between the 3rd and 4th hour of the liquefaction process. After the liquefaction and drying reactions, bio-polyols reach hydroxyl number values between 400 and 660 mgKOH/g. The heterogeneously catalyzed reaction causes the HV values to drop to a maximum of 690 mgKOH/g. Biomass conversion tells the amount of biomass that has been dissolved by the solvents. The use of sulfuric acid (VI) resulted in a biomass liquefaction of 82–96%, depending on the process temperature. The higher the process temperature, the BC reaches higher values. Using a homogeneous catalyst, a maximum of 10% cellulose solution was achieved. Other researchers liquefied biomass using various heterogeneous acids, i.e., sulfuric acid VI, hydrochloric acid, and orthophosphoric acid [30]. After a two-hour process, the researchers managed to liquefy between 55 and 75 percent of the biomass. Of the catalysts tested, sulfuric acid proved to be the best, followed by hydrochloric acid. Orthophosphoric acid had the lowest liquefaction efficiency. A slight decrease in hydroxyl number with a small BC confirms that solvent condensation occurs during liquefaction. Evidence of solvent condensation is also provided via the water content determined after the liquefaction process. Figure 3 shows the values of the water content determinations using Karl–Fisher titration (experimental) and the percentage of liquid collected in the receiver.

Water in the biomass liquefaction reaction is a byproduct and indicates the progress of the reaction. As the temperature of the process increases, more water is observed in the bio-polyol. During liquefaction with sulfuric acid (VI), the amount of water formed is significantly higher, compared to the reactions using Nafion. Figure 3 also shows the weight % values of the distillate collected. Reactions catalyzed by sulfuric acid (VI) contributed more distillate than labeled water. This is evidence that under liquefaction, there is evaporation (distillation) of solvents or/and small-molecule reaction products. It is clear that the amount of distillate is more than twice as much as the designated amount of water. In the case of reactions with Nafion, it is noticeable that the amount of labeled water is greater than the amount of distillate. In reactions with a heterogeneous catalyst, much less solution was collected in the receiver than the labeled water in the bio-polyols. The values of labeled water were very small in the range of 0.5–1.8%. As mentioned above, water is formed during liquefaction, which partially evaporates and condenses in the receiver (distillate amount). As proven in previous studies [26], measurements of the rheological properties of the obtained polyols made it possible to determine their viscosity and to fit the appropriate mathematical models describing the nature of the liquids. The results are shown in Figure 4 and Table 2.

The viscosity test was performed at 30 °C for all bio-polyols except 170_H_2_SO_4_. Due to the high viscosity of this bio-polyol, the test was performed at 60 °C. The viscosity of the obtained bio-polyols increases with the increasing temperature of the liquefaction process. The viscosity of the obtained bio-polyols using sulfuric acid VI is in the range of 1.5–13 Pas. From an application point of view, the best viscosity was achieved for the bio-polyol obtained using sulfuric acid VI at 150 °C. Other researchers have also obtained similar viscosity values for cellulose liquefaction products [31,32]. In the case of Nafion-catalyzed reactions, the viscosity values of bio-polyols are significantly lower than those of the compounds obtained with sulfuric acid (VI). The increase in viscosity indicates the formation of compounds of higher molecular weight. Bio-polyols obtained with Nafion have viscosities similar to those of solvents, indicating the absence of a reaction. The viscosity results, when compared with the hydroxyl number and biomass conversion values, confirm that solvent condensation occurs in addition to biomass liquefaction. This is particularly true for bio-polyol 170_H_2_SO_4_, where the viscosity was significantly higher than for bio-polyol 150_H_2_SO_4_ at similar biomass conversion values. Solvent condensation has been confirmed in other works [12,33]. The use of higher liquefaction temperature may have contributed to the formation of the branched and cyclic structures of glycerol or PEG400, resulting in reduced chain mobility increasing the viscosity [34]. By correlating the viscosity results with the liquid content in the receiver, it can be concluded that the evaporation of low-molecular-weight volatile compounds may have also contributed to the increased viscosity of the tested bio-polyols, as confirmed in another study [33]. To determine the nature of the liquids, Table 2 presents the data on the rheological analysis of the obtained bio-polyols, which was plotted using the Hershel–Bulkley model. The mathematical models approximate the actual behavior of non-Newtonian fluids. The Herschel–Bulkley model is described by the following relation (3):(3)$$\tau =\tau_{0}+K\gamma^{n}$$
where τ is the shear stress, $\tau_{0}$ 0 the yield stress, γ the shear rate, *K* the flow consistency index, and *n* the flow behavior index.

Knowing the value of the fluid’s flow exponent, the nature of the fluid can be determined: n < 1—pseudoplastic, n = 1—Newtonian, n > 1—dilatant. Newtonian fluid is characterized by a constant viscosity at different shear rates. In the case of pseudoplastic fluids, an increase in shear rate causes a decrease in viscosity, and in the case of dilatant fluids, an increase in viscosity. Therefore, pseudoplastic fluids are often called shear-thinning fluids, and dilatant fluids are called shear-thickening fluids [35]. All bio-polyols exhibit a non-Newtonian character. The flow exponent for all bio-polyols, except 170_H_2_SO_4_, is greater than 1, i.e., they show the nature of shear-thickened liquids, also known as dilatant liquids. Bio-polyol 170_H_2_SO_4_ shows a value of n < 1, i.e., it is a pseudoplastic liquid.

### 3.2. Chemical Structure

Fourier transform infrared spectroscopy (FTIR) and nuclear magnetic resonance (NMR) were performed to determine the chemical structure of the resulting bio-polyols. Figure 5 shows the infrared spectra for the bio-polyols.

Absorbance bands located at a wavelength of about 3300 cm^−1^ are characteristic of the -OH group stretching vibrations [36]. Hydroxyl groups are present in both solvents, cellulose, and reaction products. According to the proposed course of the main reaction in the liquefaction process (Figure 1), the product is levulinate esters. For bio-polyols obtained with sulfuric acid (VI), a reduction in the intensity of this band is observed. This is consistent with the result of determining the content of hydroxyl groups. The higher the temperature of the liquefaction process, the lower the hydroxyl number, i.e., the lower the concentration of hydroxyl groups in the molecule. At a wave number of 2800–3000 cm^−1^, absorbance bands of the stretching vibrations of -CH and -CH_2_ groups are observed [37]. For the bio-polyols obtained using the homogeneous catalyst, absorbance bands are observed at a wave number of 1700 and 1600 cm^−1^. Bands in this range are not present for the solvent mixture. They testify to the presence of the C=O and C=C groups present in the cellulose degradation products [38]. The presence of bands responsible for the stretching vibrations of C=O and C=C groups can also confirm the presence of levulinate esters and formic acid esters [39]. Signals in the wave number range of 1300 and 1360 cm^−1^ can be attributed to the bending vibrations of -CH_2_ and -CH groups [40]. Absorbance bands between 1200 and 970 cm^−1^ are characteristic of ether group vibrations [41]. Other signals below the 900 cm^−1^ wave number can be attributed to the vibrations of the -CH group in the aromatic rings of the cellulose used [42]. For bio-polyols obtained with Nafion as the catalyst, spectra similar to those of the solvents are observed. To a small extent, the spectra differ from each other, mainly in the intensity of the absorbance band responsible for the vibration of the -OH group. This confirms the conclusion that during liquefaction with Nafion, the condensation of solvents took place as a result of the concentration of hydroxyl groups decreasing, but the cellulose was not liquefied. Figure 6 shows the ^1^H NMR spectra of the bio-polyols. Depending on the catalyst used and the temperature at which the liquefaction process was carried out, the resulting bio-polyols differ in chemical structure. For bio-polyols obtained with Nafion, clear signals are observed in the 4.5 ppm region. For the other compounds, signals are also observed in this region, but they are much weaker. In the 4.5 ppm region, the protons are from carbohydrates and esters resonate [43]. In the chemical shift range of 3.0 to 4.0, solvent-derived signals are observed, mainly from protons at main-chain carbons [44,45]. According to the proposed course of the liquefaction reaction, one of the products is levulinic acid esters. Bio-polyols obtained using sulfuric acid are characterized by the presence of signals in the region of about 2.0 ppm, which is a confirmation of obtaining levulinate [25].

A ^13^C MNR study was also performed for the bio-polyols (Figure 7). In all cases, signals are observed at 70.2 and 60.6 ppm from the primary and secondary carbon atoms present in poly(ethylene glycol). Signals at 73.3 and 63.5 ppm from the primary and secondary carbon atoms derived from glycerol are also observed [46]. Peaks in this range are not observed for the 170_H_2_SO_4_ bio-polyol, which may indicate that the solvent (glycerol) has been completely reacted. In addition, peaks at 72.8 ppm from cellulose and its derivatives are observed in all bio-polyols [25].

Molecular weight measurements were made for the samples obtained using sulfuric acid VI. The results are shown in Figure 8. The average molecular weight is 1503, 3507, and 535,387 g/mol for the bio-polyols obtained at 130, 150, and 170 °C, respectively. Similar results were obtained by other researchers [47,48]. An increase in the average molecular weight is observed with an increase in the temperature of the liquefaction process, confirming the occurrence of a reaction between solvents and cellulose. The bio-polyol obtained at 170 °C has a significantly higher Mw than the other bio-polyols. This may be due to the condensation of poly(ethylene glycol), which can polycondense to a mass of 10,000,000 g/mol [49].

### 3.3. Thermal Properties

Thermogravimetric analysis coupled to a gas analyzer (FTIR) and differential scanning calorimetry were used to determine the thermal properties of the bio-polyols by examining the oxidation onset temperature (OOT). The results are shown in Table 3. For all bio-polyols, a higher liquefaction temperature increased the thermal stability. The bio-polyols obtained using sulfuric acid (VI) as a catalyst had a significantly higher thermal stability. For these bio-polyols, the thermal stability is in the range of 148.5–177.0 °C and is significantly higher than that of the solvents. The increased thermal stability indicates the formation of higher molecular weight compounds. Bio-polyols obtained with Nafion as the catalyst showed lower thermal stability than the solvents. The reason for this may be the unfluidized cellulose, which, due to its lower thermal stability than the solvents, contributed to the lower stability of the bio-polyols. Analyzing the DTG curves, it can be seen that all bio-polyols degrade in two stages. The first degradation step at T_max1_ = 210 °C can be attributed to the degradation of glycerol and its derivatives [49]. The second stage of degradation around T_max2_ = 340 °C can be attributed to the degradation of PEG400 chains, its derivatives, and oligomeric compounds formed via solvent condensation (e.g., polyglycerol) and liquefaction (e.g., levulinates) reactions. 

Table 3 also shows the results of the OOT analysis. The oxidation onset temperature was defined as the intersection of the tangent line to the initial baseline and the tangent line to the point of maximum increase, corresponding to the inflection point of the oxidation peak. For all bio-polyols, as the liquefaction temperature increases, the OOT value increases. Nevertheless, it can be noted that bio-polyols obtained using sulfuric acid (VI) as a catalyst have higher OOT values. Comparing the OOT values of bio-polyols and the solvent mixture (GLY_PEG400), two relationships are observed. First, the use of Nafion as a catalyst resulted in a lower oxidation onset temperature, which may be due to solvent condensation. This has been confirmed in previous studies [47]. Secondly, sulfuric acid (VI) as a catalyst resulted in an increase in the degradation onset temperature, which may confirm the cellulose liquefaction reaction [26]. In order to study the gaseous products of the thermal decomposition of bio-polyols, TGA-FTIR analysis was carried out and the results are shown in Figure 9.

In the wavelength number range of 3400–4000 cm^−1^ and 1250–2000 cm^−1^, absorbance bands associated with water vapor are observed [50,51]. The peak maxima observed in the 2900–2950 cm^−1^ range are associated with symmetric and asymmetric vibrations of the -CH, CH2, and CH3 groups present in the aliphatic chains. The bending deformation vibrations of these groups show absorbance bands in the range of about 1000 cm^−1^. In the range of 2300, 2350–2370 cm^−1^ signals related to the stretching of double bonds in the CO_2_ molecule are observed. Absorbance bands observed at 1710–1780 cm^−1^ are associated with the stretching vibrations of -C=O bonds. In the 1690–1745 cm^−1^ range, signals are observed from the stretching of the C=C alkene bonds. The absorbance peak at 1100 cm^−1^ is characteristic of the stretching vibrations of the C-O group. In summary, the main products of thermal decomposition of the obtained bio-polyols regardless of the type of catalyst used include water, carbon dioxide, aliphatic hydrocarbons, and compounds containing a carbonyl group, such as aldehydes and alcohols.

## 4. Conclusions

This paper presents a comparison of the efficiency of cellulose liquefaction using two different catalysts. The heterogeneous catalyst was Nafion ion-exchange resin and the homogeneous catalyst was the commonly used sulfuric acid (VI). Six different bio-polyols were obtained using two types of catalysts and three different liquefaction process temperatures of 130, 150, and 170 °C. The effects of the mentioned conditions on the physicochemical, rheological, and thermal properties and the chemical structure of the obtained compounds were determined. Based on the values of the hydroxyl number and biomass conversion, it was found that higher liquefaction efficiency was obtained using sulfuric acid (VI) as a catalyst. The hydroxyl number values for these bio-polyols were lower and biomass covariance was higher, compared to the bio-polyols obtained using Nafion. The bio-polyol 170_H_2_SO_4_ had the lowest hydroxyl number and the highest biomass conversion. At the same time, it showed the highest viscosity (13.49 Pas at 60 °C). Such high viscosity precludes its use in polyurethane processing. Rheological studies showed that all bio-polyols exhibit the properties of a non-Newtonian fluid, thickened by shear. The viscosity of the compounds tested at 30 °C was in the range of 0.108–3.03 Pas. Chemical structure analysis using FTIR and 1H NMR confirmed the presence of levulinic acid esters. The thermal stability of bio-polyols obtained using sulfuric acid (VI) was higher compared to that of the solvents, indicating the formation of higher molecular weight compounds in the reaction scorer. The OOT study confirmed that increasing the temperature of the liquefaction process increases the oxidative stability of bio-polyols. Increased thermal and oxidative stability may benefit potential applications in the synthesis of polyurethanes. A scheme of possible chemical reactions occurring during liquefaction is also presented. It was determined that in addition to the liquefaction of cellulose, the condensation of solvents also occurs. It was also confirmed that the liquefaction of biomass requires a catalyst that will be a suitable proton donor. Nafion ion-exchange resin was found to be ineffective in the liquefaction process. The reason may be the poor reaction environment; ion-exchange resins often activate in aqueous environments. Although water is a byproduct during liquefaction, there may have been too little water to activate the catalyst.

## Figures and Tables

**Figure 1 materials-16-06135-f001:**
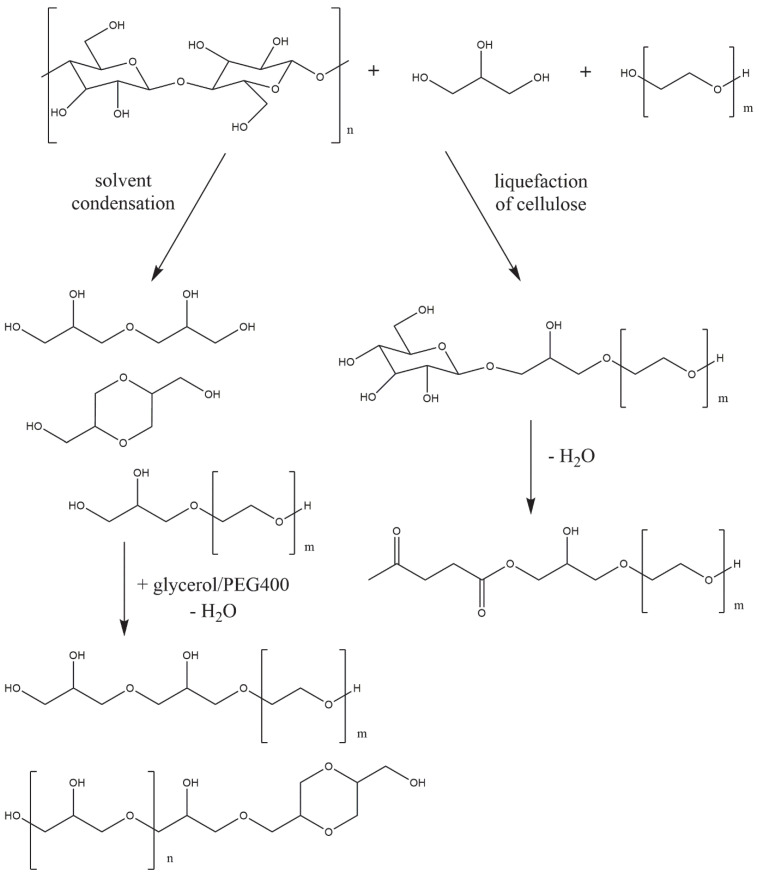
Possible reactions occurring during cellulose liquefaction.

**Figure 2 materials-16-06135-f002:**
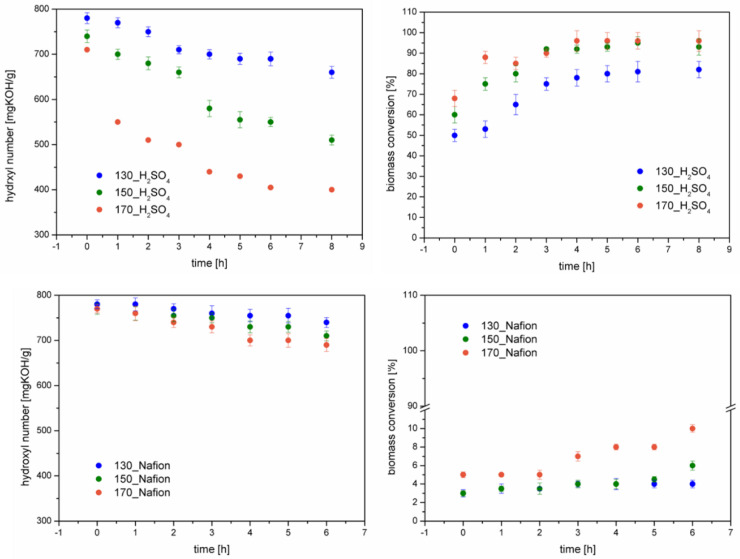
Changes in hydroxyl number and biomass conversion during the liquefaction process ((**Upper charts**)—homogeneous catalyst, (**lower charts**)—heterogeneous catalyst).

**Figure 3 materials-16-06135-f003:**
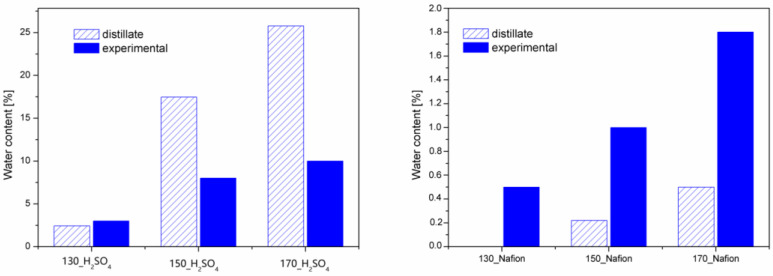
Water content and the amount of liquid in the receiver ((**Left diagram**)—homogeneous catalyst, (**right diagram**)—heterogeneous catalyst).

**Figure 4 materials-16-06135-f004:**
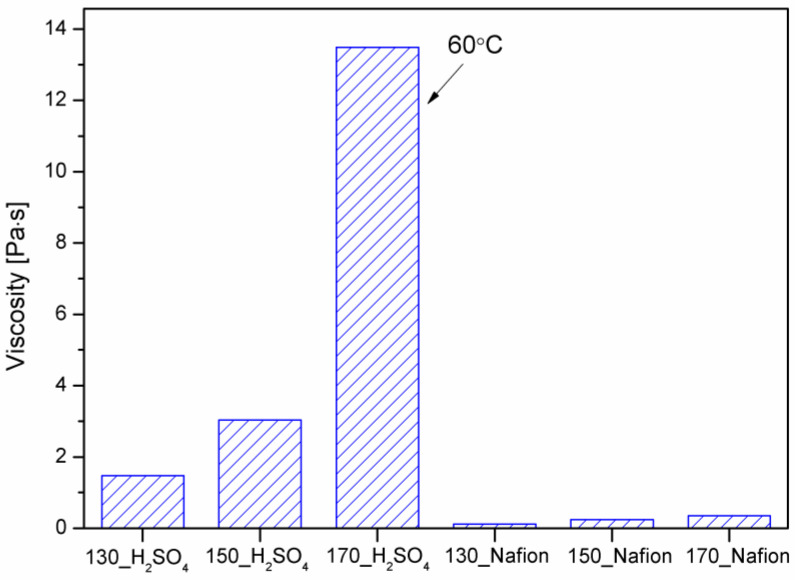
Viscosity of bio-polyols.

**Figure 5 materials-16-06135-f005:**
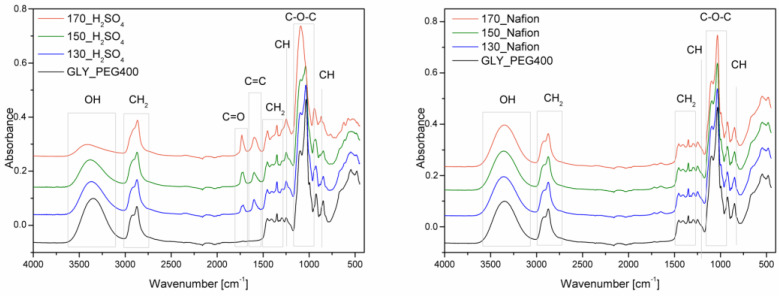
Infrared spectroscopy spectra of bio-polyols ((**Left diagram**)—homogeneous catalyst, (**right diagram**)—heterogeneous catalyst).

**Figure 6 materials-16-06135-f006:**
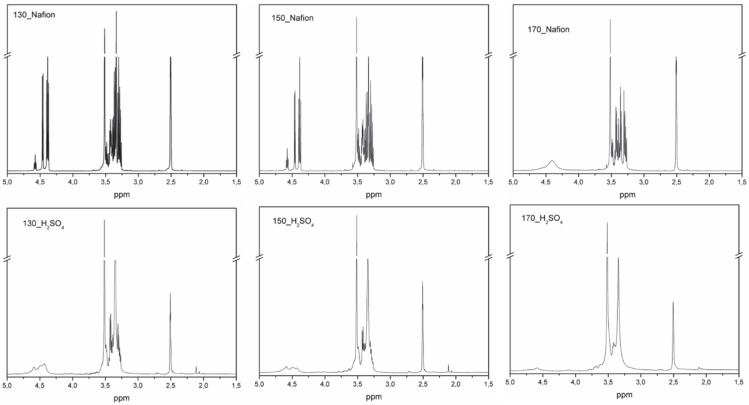
^1^H NMR spectrum of bio-polyols.

**Figure 7 materials-16-06135-f007:**
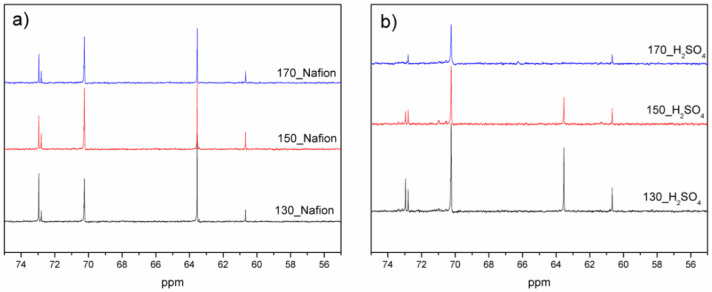
^13^C NMR spectrum of bio-polyols. (**a**) heterogeneous catalyst, (**b**) homogeneous catalyst.

**Figure 8 materials-16-06135-f008:**
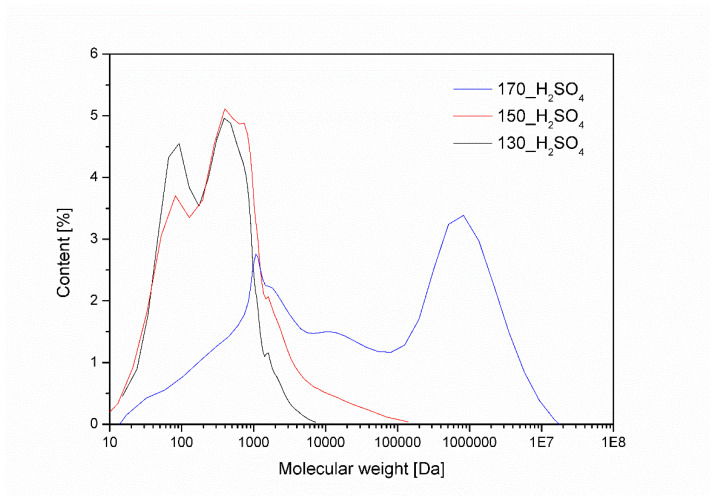
Molecular weight of the obtained bio-polyols using sulfuric acid (VI) as a catalyst.

**Figure 9 materials-16-06135-f009:**
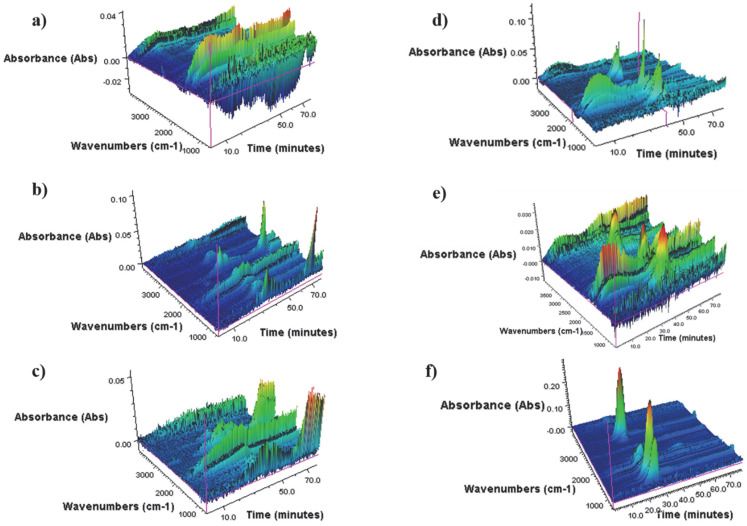
3D FTIR spectra of degradation products of bio-polyols: (**a**) 130_H_2_SO_4_, (**b**) 150_H_2_SO_4_, (**c**) 170_H_2_SO_4_, (**d**) 130_Nafion, (**e**) 150_Nafion, and (**f**) 170_Nafion.

**Table 1 materials-16-06135-t001:** The components used in the synthesis of bio-polyols.

Sample Name	Component [g]				
	Cellulose	Glicerol	PEG400	Catalyst	NaOH
130_H_2_SO_4_	40	200	200	12	9.7
150_H_2_SO_4_	40	200	200	12	9.7
170_H_2_SO_4_	40	200	200	12	9.7
130_Nafion	40	200	200	24	-
150_Nafion	40	200	200	24	-
170_Nafion	40	200	200	24	-

**Table 2 materials-16-06135-t002:** The Herschel–Bulkley linear functions.

Sample Code	Function	τ_0_ [Pa]	K [Pa*s^n^]	n [-]	R^2^
130_H_2_SO_4_	y = 1.4039 * x^1.0142^	0	1.4039	1.0142	0.9999
150_H_2_SO_4_	y = 2.9876 * x^1.0048^	0	2.9876	1.0048	0.9999
170_H_2_SO_4_	y = 15.2534 * x^0.9758^	0	15.2534	0.9758	0.9999
130_Nafion	y = 0.0227 * x^.1.3470^	0	0.0227	1.3470	0.9954
150_Nafion	y = 0.1086 * x^.1.1732^	0	0.1086	1.1732	0.9970
170_Nafion	y = 0.3170 * x^.1.0182^	0	0.3170	1.0182	0.9952

**Table 3 materials-16-06135-t003:** Thermal properties of bio-polyols.

	Temperature [°C]				
	T_2%_	T_5%_	T_max1_	T_max2_	OOT
GLY_PEG400	133.4	162.0	213.1	313.5	194.2
130_H_2_SO_4_	148.5	172.4	211.3	346.6	193.2
150_H_2_SO_4_	154.6	178.7	211.4	334.6	198.8
170_H_2_SO_4_	177.0	206.2	300.2	373.9	228.2
130_Nafion	93.8	155.7	209.2	325.5	170.0
150_Nafion	106.8	153.1	208.8	347.3	185.1
170_Nafion	125.5	157.6	212.9	335.9	198.5

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
