# Peer review of "Comparison of the Efficiency of Hetero- and Homogeneous Catalysts in Cellulose Liquefaction"

_materials, 2023, doi:10.3390/ma16186135_

Round 1

Reviewer 1 Report (Previous Reviewer 2)

The work present a comparison of catalyst for cellulose liquefaction. Overall the manuscript content is acceptable. Some comments for improvement.

1. The reasoning of specifying the usage of that specific 2 types of catalyst needs to be further clarified and justified. Has the catalyst been used in others reaction? what is the superiority in performance? Compare it with existing and common catalyst to justify its use. The reason of 'it has not been use before' is valid but insufficient.

2. For figure 3, the column present water content and liquid mixture? Water would be the less required product here i presumed. Suggest the author to conduct Karl fisher titration method to identify the total water content, hence able to obtain the exact amount of liquid. A better mass balance can be observed. 

3. Are there still plenty of water since drying has been done prior to  experiment, if so, where does the water comes ffrom?

4. label all figures properly, there are at least 2 different pictures in a single figure, lable them (a), (b), etc and describe in the caption

5. The bio-polyols obtained comes in different variety, what is the used? will it required extraction or they can be treated as a single product? 

acceptable

Author Response

The responses for the Reviewer are located in the attachment.

Reviewer 2 Report (Previous Reviewer 1)

This paper compared two different catalysts (heterogeneous Nafion ion-exchange resin and homogeneous sulfuric acid) and their cellulose liquefaction efficiency at different temperatures. By analyzing the obtained bio-polyols products, new information was provided on the possibility of using heterogeneous catalysts in cellulose liquefaction. This reviewer recommends publishing this paper with the consideration of the following in the revision.

1.      Compared to other studies, especially in recent research publications, the author needs to elaborate on the scientific novelty of this manuscript.

2.      Based on the characterization results, the author claimed that higher liquefaction efficiency was obtained using sulfuric acid catalyst and suggested applying granular potassium hydroxide to neutralize the obtained bio-polyols. Nevertheless, it is necessary to carefully consider the reduction in economy and difficulties in industrial application caused by direct neutralization with alkali.

Author Response

The responses for the reviewer are located in the attachment.

This manuscript is a resubmission of an earlier submission. The following is a list of the peer review reports and author responses from that submission.

Round 1

Reviewer 1 Report

Biomass liquefaction is important for industrial utilization of biomass resouces.  This manuscript demonstrates the effectiveness of a heterogeneous Nafion ion-exchange resin on the cellulose liquefaction process compared to the commonly using sulfuric acid. The research provides new information on the possibility of using heterogeneous catalysts in cellulose liquefaction. Before publication, there are some issues to be addressed.

1. For products analysis, gel permeation chromatography (GPC) and 13C-NMR are recommended. 

2. Homogeneous sulfuric acid has been proved higher efficiency than heterogeneous Nafion. However, seperating catalysts from products is a predictable issue in this case.  Therefore, more discussion about liquid product purification should be supplied. 

3. Results should be comparied with more reported catalysts. 

Author Response

Thank you for the review. I am sending the response in the attachment.

Reviewer 2 Report

The work compared 2 types of catalyst from hetero- and homogeneous group. The experimental method are a common approach and the novelty of this work is claimed to be the use of Nafion NR40. There are several others types of catalyst available but not considered in this work, where I personally feel that the current scope is insufficient. I suggest the author could expand the scope of cover at least 4 types of catalyst- 2 of each group for a better and robust comparison. I would recommend the author to resubmit since additional experimental studies is needed. Additional comments to improve the work as follows:
1.       xonessential’ what does this term means?
2.       There have been heterogeneous catalysts being mentioned in the literature section but not Nafion NR40. The author claims that this has not been studied before, what are the speciality of this catalyst as compared to those studied before that prompt the author to study it?
3.       Why the author did not include more catalyst in the study since this is a comparison study. Merely comparison just 2 types doesn’t give a sufficient research scopes.
4.       Why the amount of catalyst differs between both types?
5.       The author proposed a possible reaction pathway in Figure 1. Any method/ analysis done to confirm any of these steps, or are they merely just guess?

6.       Why the data collection for both catalyst is inconsistent? 7hr for Na and 9hr for acid. And the data for the 7th hour for acid catalyst is missing.

English quality is acceptable. 

Author Response

(The authors gave the same response as above.)

Reviewer 3 Report

The purpose of the present study is to test the effectiveness of a heterogeneous catalyst such as Nafion ion-exchange resin on the cellulose liquefaction process. The results obtained were compared with bio-polyols obtained in a conventional way, using a homogeneous catalyst (sulphuric acid).   The research provides new information on the possibility of using heterogeneous catalysts in cellulose liquefaction.  The paper could be published after revision.

-Introduction of the paper is long. It should be more concentrated.

- Cellulose was liquefied using solvents in the form of a mixture of glycerol and PEG400 in a 50/50 mass ratio. Why such ratio of the solvents was chosen?

-The amount of catalyst was 3 (homogeneous catalyst) or 6% w/w. (heterogeneous catalyst) in relation to the solvents. Why the different amounts were used. The same amounts should be tested for exact comparison.

- The highest temperature in the process was 170°C. Could a higher temperature be used?

-Physicochemical, rheological, thermal properties and chemical structure of the obtained compounds should be also compared in conclusions with those of other cellulose liquefaction products, which are described in literature.

Author Response

(The authors gave the same response as above.)

Round 2

Reviewer 2 Report

There is no additional experiment done. The scopes of experiment is very limited. The author's replies from previous comments are not convincing. By stating 'additional experiment will be done in the future' doesnt add-in value to the current work. 

Reviewer 3 Report

Accept in present form